# Spontaneous Imbibition Oil Recovery by Natural Surfactant/Nanofluid: An Experimental and Theoretical Study

**DOI:** 10.3390/nano12203563

**Published:** 2022-10-12

**Authors:** Reza Khoramian, Riyaz Kharrat, Peyman Pourafshary, Saeed Golshokooh, Fatemeh Hashemi

**Affiliations:** 1School of Mining and Geosciences, Nazarbayev University, Astana 010000, Kazakhstan; 2Department Petroleum Engineering, Montanuniversität, 8700 Leoben, Austria; 3Faculty of Petroleum and Natural Gas Engineering, Sahand University of Technology, Tabriz 513351996, Iran; 4Faculty of Chemistry, Shiraz University, Shiraz 7155713876, Iran

**Keywords:** natural surfactant, nanoparticles, spontaneous imbibition, mathematical modeling, enhanced oil recovery

## Abstract

Organic surfactants have been utilized with different nanoparticles in enhanced oil recovery (EOR) operations due to the synergic mechanisms of nanofluid stabilization, wettability alteration, and oil-water interfacial tension reduction. However, investment and environmental issues are the main concerns to make the operation more practical. The present study introduces a natural and cost-effective surfactant named Azarboo for modifying the surface traits of silica nanoparticles for more efficient EOR. Surface-modified nanoparticles were synthesized by conjugating negatively charged Azarboo surfactant on positively charged amino-treated silica nanoparticles. The effect of the hybrid application of the natural surfactant and amine-modified silica nanoparticles was investigated by analysis of wettability alteration. Amine-surfactant-functionalized silica nanoparticles were found to be more effective than typical nanoparticles. Amott cell experiments showed maximum imbibition oil recovery after nine days of treatment with amine-surfactant-modified nanoparticles and fifteen days of treatment with amine-modified nanoparticles. This finding confirmed the superior potential of amine-surfactant-modified silica nanoparticles compared to amine-modified silica nanoparticles. Modeling showed that amine surfactant-treated SiO_2_ could change wettability from strongly oil-wet to almost strongly water-wet. In the case of amine-treated silica nanoparticles, a strongly water-wet condition was not achieved. Oil displacement experiments confirmed the better performance of amine-surfactant-treated SiO_2_ nanoparticles compared to amine-treated SiO_2_ by improving oil recovery by 15%. Overall, a synergistic effect between Azarboo surfactant and amine-modified silica nanoparticles led to wettability alteration and higher oil recovery.

## 1. Introduction

Oil recoveries have declined in many oil fields worldwide [1]. Different chemical and physical techniques have improved oil recovery [2,3]. Wettability alteration is an effective mechanism that results in higher oil recovery [4,5]. Nanoparticles are proposed as efficient wettability modifiers during oil extraction [6]. Destabilization under harsh conditions of reservoirs has led researchers to modify nanoparticles with different surfactant agents [7].

Even though the role of surfactants in oil reservoirs is mainly to modify the wetting condition of oil-wet rocks and reduce the interfacial tension (IFT), they can also disperse or stabilize nanoparticles [8]. Nwidee et al. [9] showed that the functional ZrO_2_ nanoparticles facilitated wettability alteration by adsorption on the rock surface, confirmed by microscopic images and contact angle measurements. Imbibition tests revealed a fast water imbibition process for the rock samples coated with the surfactant-modified nanofluid. Rezk and Allam [10] unveiled the synergetic effect of sodium dodecylbenzene sulfonate (SDBS) anionic surfactant and zinc oxide (ZnO) nanoparticles on interfacial tension, wettability, and oil productivity. A remarkable decrease in the interfacial tension was observed upon adding ZnO nanoparticles into the surfactant solution and attributed to the nanoparticles’ low polydispersity and uniformity. The ZnO-based nanofluid, overcoming the capillary pressure, altered wettability to further water wet state and improved oil recovery by 8%. Zhao et al. [11] combined a nonionic surfactant and SiO_2_ nanoparticles for EOR applications. Imbibition studies showed higher oil recovery than the standalone application of nanofluid or surfactant. Soleimani et al. [12] synthesized ZnO nanoparticles via the sol-gel method and dispersed them in the aqueous phase using sodium dodecyl sulfate (SDS). The highest oil recovery was observed at 3000 ppm ZnO due to interfacial tension and wettability alteration. Divandari et al. [13] coated magnetic nanoparticles with a surfactant and proved better wettability alteration and lower precipitation.

Cetyltrimethylammonium bromide (CTAB) is a typical cationic surfactant with long-chain carbons, which bears positive charges on the polar portion of fluids [14]. Ma et al. [15] studied surface modification of silica nanoparticles with CTAB nanoparticles at different temperatures and concentrations. Due to the positive charges of CTAB and negative charges of SiO_2_, the CTAB could be absorbed on the surface of nanoparticles and improve the dispersal state of the nanoparticles. Using sand column experiments, Liu et al. [16] observed better transport and retention of graphene oxide nanosheets dispersed in CTAB and SDBS. Panahpoori et al. [17] improved CTAB foam stability at harsh reservoir conditions using TiO_2_ nanoparticles. Pereira et al. [18] modified the surface of Fe_3_O_4_ nanoparticles using CTAB. The resultant nanoparticles were stable even in divalent cations and more capable of altering the rock wettability. Joshi et al. [19] investigated using SiO_2_ with polymers and CTAB surfactants to increase the oil recovery from oil reservoirs. The stability of nanofluids was improved when surfactant agents were utilized. Synergistic effects of polymer, nanoparticle, and surfactant contributed to IFT reduction, wettability alteration, and viscosity enhancement. Hethnawi et al. [20] studied the interfacial behavior of CTAB-grafted faujasite-based nanoparticles under various conditions. The developed nanofluid showed a considerable improvement in IFT and viscoelasticity.

The published EOR studies have focused on nanofluids combined with synthetic surfactants like CTAB, SDBS, and SDS, which are known to be toxic aquatic organisms [21]. Recent international regulations prohibit using non-biodegradable and toxic chemicals [22], which contributes to phasing out some surfactant agents. To address this challenge, researchers’ focus has shifted to new non-toxic alternatives for petrochemical surfactants. Nowrouzi et al. [23] prepared a non-petrochemical surfactant from powder leaves of Myrtus communis, a source of natural surfactants. The surfactant increased oil recovery by 14.3% and reduced IFT to 0.86 mN/m and had low adsorption on the rock surface. Khayati et al. [24] found pure saponin very effective for IFT reduction and wettability alteration to hydrophilicity. Emadi et al. [25] investigated the impact of foam generated by Cedr extract on mobility control and introduced it as an advisable chemical agent for EOR. Pa et al. [26] fabricated sunflower Gemini surfactants, leading to stable emulsions, ultralow IFT, and great foamability. Traiwiriyawong et al. [27] extracted a benign surfactant from palm kernel oil and used it in wettability studies. It showed the least adsorption compared to commercial surfactants of SDS and CTAB.

Recently, amine molecules have been used for surface modification of nanoparticles rather than chemical surfactants. Wang et al. [28] aminated SiO_2_ nanoparticles with tris(hydroxymethyl)aminomethane and steric acid to increase hydrophobicity. The contact angle of SiO_2_ was initially around 18° due to several hydroxyl groups on the SiO_2_ surface, but it increased by almost 100° after amination, implying high hydrophilicity of animated nanoparticles. Habibi et al. [29] utilized amines and organsiloxane for homogenized dispersibility and surface modification of SiO_2_ nanoparticles. The surface modification remarkably improved wettability and surface activity, resulting in higher oil recovery in micromodel floodings. In another study [30], aminobutanol was utilized to improve the surface activity of SiO_2_ nanoparticles by amination. The reactivity of the nanoparticles was increased and enabled them to be grafted more easily to carboxylic acids.

This study amins to aminate SiO_2_ nanoparticles and combine them with a new green natural surfactant from bio-sources called Azarboo (Chooback). Hence, SiO_2_ nanoparticles are first modified with positively charged amine groups to become ready to absorb Azarboo anionic surfactant (Figure 1). The performance of this new chemical is experimentally studied in this work. The experimental results are then analyzed using the capillary number and validated by the analytical approach by Mattax and Kyte [31], Ma et al. [32], and Aronofsky et al. [33].

## 2. Experimental Section

### 2.1. Materials

The natural surfactant powder of Azarboo (Chooback) was utilized in this study. This green surfactant was extracted from hard and bony roots of *Acanthophyllum*, which has a bitter taste and yellow color. Due to having hydrophilic and hydrophobic parts at the same time, it can create thick and stable foams [34].

Hydrophilic colloidal silicon oxide nanoparticles were utilized in this investigation with a purity of >99.9 wt.%. The mean particle size of the nanomaterials was between 5 and 15 nm. Different chemical and physical characteristics of the nanoparticles are shown in Table 1.

Carbonate core samples of 3.8 cm in diameter and 6.5 cm in length were utilized in wettability alteration and core flooding experiments. Table 2 shows the properties of core samples. Two slices with 3 mm thickness were cut from one of the core plugs and polished to be smooth enough for contact angle experiments.

Ethanol (99%) and 3-aminopropyltriethoxysilanec (APTS) were from Merck Company (Darmstadt, Germany). The chemical composition and properties of the degassed oil used in this study are also listed in Table 3.

### 2.2. Methodology

#### 2.2.1. Amine Functionalization

Amine functionalized silica nanoparticles were prepared using the reaction of silica nanoparticles and APTS at room temperature. 1 mL APTS was dissolved into 200 mL ethanol in a beaker on a stirrer. Then, 10 g silica nanoparticle was added to the beaker and sonicated for 1 h. It was followed by adding 150 mL distilled water to the mixture and sonication for half an hour. The mixture was centrifuged under 15,000 rpm for almost 20 min, and the precipitated part was collected and washed with ethanol. Amino-modified silica nanoparticles (Si-NH_2_) were obtained by gently heating the gel-like precipitation at 50 °C for 6 h [35].

#### 2.2.2. Natural Surfactant Extraction and Optimization

The maceration procedure [36] was used to get the *Acanthophyllum* plant extract. The bony roots of this plant were dried at ambient temperature and pulverized using an electric mortar and pestle. Then, almost 400 g of the dried plant powder was combined with distilled and kept in a sealed Erlenmeyer flask for at least three days. The Erlenmeyer flask was shaken using an orbital shaker to mix the powder with water continually. The flask’s contents were then filtrated and transferred into a digital rotary evaporator flask (DLAB) for about five hours to obtain a dry extract powder (Figure 2).

Surfactant solutions were prepared at different concentrations (200–2000 ppm) by combining the extracted Azarboo and synthetic brine (180,000 ppm NaCl) on a magnetic stirrer for 10 min. Then, electrical conductivity and IFT were measured to find the surfactant’s critical micelle concentration (CMC). The IFT and conductivity values were recorded versus concentration (Figure 3), and CMC was found at 1200 ppm. No further tests were done beyond this threshold, as the IFT does not change after the CMC value [37,38].

Measurements showed an almost 39% decrease in IFT (Figure 3), which confirmed a very good influence of the natural surfactant on emulsification and IFT reduction, which is considered as effective EOR mechanisms.

#### 2.2.3. Surfactant/Nanofluid Preparation

It has been proven that nanoparticles can penetrate the micron-sized pores and throats of the reservoir rocks and improve oil recovery afterward [39,40]. The nanoparticles can unfavorably affect fluid flow in the porous media if the nanofluid concentration exceeds a certain amount due to entrapment [41,42]. Hence, a concentration of 500 ppm of the Si-NH_2_ nanoparticles was selected and applied to prepare Azarboo/nanofluid. The concentration had been introduced by extensive research for various nanoparticles [43,44]. The Azarboo/nanofluid was prepared by dispersing 500 ppm of the modified nanoparticles into the surfactant solution (1200 ppm surfactant with 180,000 ppm NaCl). To better evaluate the properties of the developed nanofluid, samples without the natural surfactant (pure silica and Si-NH_2_) were also prepared. The nanofluids were softly stirred for two hours (one hour with an ultrasonic probe and one hour inside an ultrasonic bath) at 20 kHz. The objective was to utterly suspend the nanoparticles into the dispersion medium and prevent aggregation.

#### 2.2.4. Surfactant Characterizations

Fourier transform infrared (FT-IR) spectrum of Azarboo was recorded on a PerkinElmer Spectrum™ 3 FT-IR spectrometer (Waltham, MA, USA) and compared with those of amine-treated and surfactant-modified nanoparticles for functional analysis. The proton nuclear magnetic resonance (H-NMR) spectroscopy was done using Bruker 500 MHz EPR (Billerica, MA, USA), to determine the structure of Azarboo molecules. The natural surfactant was thermally studied by thermogravimetric analysis (TGA). TGA, which was performed using TGA-Q600 SDT (Milford, CT, USA) is a technique to detect how materials behave when subjected to heat. Three milligrams of natural surfactant were poured into a crucible and heated to almost 350 °C at a rate of 10 °C/min under a nitrogen atmosphere [45].

The particle size distribution was measured using a dynamic light scattering (DLS) instrument from Malvern Company (Worcestershire, United Kingdom). The Brunauer-Emmett-Teller (BET) technique was utilized to measure the surface areas of silica nanoparticles before and after modification through gas adsorption analysis by a Miraesi KICT-SPA 3000 Instrument (Miraesi, Korea). Scanning electron microscopy (SEM) images were captured using an electron microscope (HITACHI, model SU7000, Tokyo, Japan) to study nanoparticles’ morphology. The average zeta potential values of the nanoparticles were measured using a zeta potential analyzer (ZEECOM ZC2000ML Microtec Company, Brixen, Italy) at 25 °C. The zeta potential values were recorded by averaging three zeta potential measurements for each sample based on previous studies [46,47]

#### 2.2.5. Oil-Wet Procedure

All samples were saturated and soaked in crude oil at 50 °C for three weeks to be oil-wet before imbibition experiments. After aging, they were found entirely oil-wet due to having contact angles lower than almost 150° and color change from gray to dark brown (Figure 4).

#### 2.2.6. Spontaneous Imbibition

The spontaneous imbibition test [48] was used to assess the wetting condition of the rocks qualitatively. Two core plugs (No. 1 and 2) with induced oleophilic wettability were drenched in the prepared nanofluids for 24 h at room temperature. Each sample was taken out and dried at 40 °C for one day. They were saturated with oil and placed inside brine-filled Amott cells at 50 °C. The volume of oil expelled was measured by monitoring the graduation of the cell [49]. Another imbibition test was conducted with an oil-wet sample without treatment with nanofluids.

Mattax and Kyte [31] proposed a scaling group for imbibition oil recovery from strongly water-wet systems with distinct rock and fluid characteristics as
(1)$$t_{D}=\left(\frac{0.00031415}{L_{C^{2}}}\sqrt{\frac{k}{\varphi }}\;\frac{\sigma_{ow}}{\sqrt{\mu_{o}\mu_{w}}}\right)t$$
where *t_D_* is a dimensionless time, *L_C_* is a characteristic length (cm), *k* is permeability (mD), *φ* is porosity, *σ_ow_* is oil-water interfacial tension (dyne/cm), *μ_o_* is oil viscosity (cp), *μ_w_* is water viscosity (cp), and *t* is the imbibition time (hr.). Ma et al. [32] developed a single-parameter model, which was a simplified form of the Aronofsky et al. [33] model as
(2)$$\frac{R}{R_{Max}}=1-e^{-\alpha t_{D}}$$
where *R* is imbibition oil recovery, *R_Max_* is ultimate oil recovery by free imbibition, and α is the decline constant of oil production.

#### 2.2.7. Core Flooding Experiments

Displacement experiments were performed using a core flood apparatus shown in Figure 5. Cores No. 3 and 4, previously aged to become oil-wet, were selected for this section. The core plugs were washed and cleaned with toluene, methanol, and distilled water using the Soxhlet extractor to remove any dirt. Then they were heated in a furnace at almost 100 °C for 24 h to be dried [50]. The cores were saturated by a brine of 180,000 ppm NaCl. Oil was injected until no additional brine was expelled from the cores and irreducible water saturation was established. After that, the synthetic brine was flooded into the cores to mimic the secondary oil recovery. One pore volume of pure Si-NH_2_ and Si-NH_2_ modified with Azarboo was then injected into the core plugs as the tertiary oil recovery stage. The injection was stopped, and the core was soaked in the nanofluids for 24 h to alter the pores’ wettability [51]. After the shut-in treatment, the cores were fully saturated with the brine and then with the oil until the irreducible water saturation was established. This stage aimed to monitor oil recovery after the nanofluid treatment. In the flooding tests, the temperature was 50 °C, the flow rate was 0.1 cc/min, and the radial confining pressure was almost 500 psi higher than the injection pressure.

## 3. Results and Discussions

### 3.1. Characterization Results

#### 3.1.1. Natural Surfactant

The functional groups and chemical compositions of the natural surfactant were studied by FT-IR, H-NMR, and TGA analyses. In the FT-IR spectrum shown in Figure 6a, the peak at 1057 cm^−1^ is associated with C–O stretching vibration, and the peak at 1323 cm^−1^ corresponds to the –OH bond [52]. The peak at 1625 cm^−1^ in the carbonyl stretching region was mainly due to a covalent bond between two carbon atoms (C=C) [34]. Also, the peak around 2900 cm^−1^ was linked to C–H aliphatic sapogenin saponin graft [53]. Commonly, intense broadband at 3000–3600 cm^−1^ area can be seen in the IR spectrum of polysaccharides [54]. This strong band, which represents the stretching vibration of multiple hydroxyl groups (–OH) in polysaccharides, was observed at 3418 cm^−1^. These characteristic functional groups exist in the structure of Azarboo surfactant.

In the ^1^H-NMR spectrum, different chemical peaks were observed as shown in Figure 6b. The peak at about 4 ppm corresponds to the hydroxylic group (-OH). The chemical shifts from 2.8 to 3.8 ppm are related to the saponin oligosaccharide functional group, and those between 1.5 and 2.5 ppm are attributed to the glycoside-free aglycone section of the saponin [34]. These results were consistent with the presence of the FT-IR bands at 1057 cm^−1^, 2900 cm^−1^, and 3418 cm^−1^, as discussed above.

TGA analysis for Azarboo surfactant was performed under a nitrogen atmosphere. As illustrated in Figure 6c, the natural surfactant was thermally fully stable up to about 75 °C, beyond which the weight loss initiated and steadily continued to 160 °C. A probable reason for that weight loss is water evaporation from molecules and particles [55]. The thermal stability of the surfactant followed the decreasing trend more steeply, reaching 300 °C, where only less than 1% of the natural surfactant remained unchanged. This heavyweight loss, which is due to carbon bond breakdown at high temperatures [56], reveals that Azarboo is natural and extracted from plants [57]. In conclusion, TGA analysis confirms the thermal usability of the Azarboo surfactant for harsh-temperature EOR operations due to great mass maintenance at temperatures below 100 °C.

#### 3.1.2. Aminated Silica Nanoparticles

FT-IR analysis was used to indicate amine modifications on SiO_2_ nanoparticles. Figure 7 shows the IR spectrum of silica nanoparticles before and after modification. Looking at the FT-IR spectroscopy of silica nanoparticles (Figure 7, red line), the bands at 779 and 1097 cm^−1^ are attributed to bending vibration or asymmetric stretching vibration of Si–O–Si bonds [58]. The absorption bands at 1583 and 3490 cm^−1^ are assigned to O–H stretching [59]. The Si-NH_2_ nanoparticles were detected through new peaks in IR spectra (Figure 7, blue line). The new bands at 1562 and 1716 cm^−1^ originate from amine groups’ N–H bending vibration [58]. Also, the broad and strong band at 3477 cm^−1^ may be attributed to the substitution of –OH stretching with the N–H stretch of amine [60]. All these observations represent that amine-functionalized silica nanoparticles have been synthesized successfully.

The zeta potential of bare and Si-NH_2_ nanoparticles was measured to confirm the N–H conjugation on the surface. Bare SiO_2_ nanoparticles were quantified as a control, which indicated a negative zeta potential (−25 mV). Contrarily, the zeta potential for Si-NH_2_ nanoparticles was positive (+21 mV). As silica nanoparticles are filled with negative charges, and amino groups are replete with positive ones, it is evident that amine functionalization has been done efficiently (see Figure 1).

#### 3.1.3. Amino-Surfactant-Modified Silica Nanoparticles

The linkage of Azarboo surfactant to Si-NH_2_ nanoparticles was investigated by FT-IR spectroscopy. The IR spectrum of the surfactant was measured as a control (Figure 6a) to identify new functional groups in amino surfactant nanocomposite (Si–NH_2_-surfactant) after surfactant modification (Figure 8a). The results showed that all functional groups observed in Azarboo surfactant molecules appeared in IR spectroscopy of Si–NH_2_-surfactant. In addition, a weak peak at 792 cm^−1^ showed the bending vibration of Si–O–Si bonds, and a sharp peak at 1723 cm^−1^ represented the vibration of N–H. As evidenced in Figure 8, it is proven that Azarboo surfactants are linked to Si-NH_2_ nanoparticles.

The zeta potential of Si-NH_2_-surfactant nanoparticles was also recorded. The zeta potential reached a negative value (−17 mV) from a positive value (+21 mV). It confirmed surfactant conjugation on positively charged aminated silica nanoparticles (see Figure 1). In addition, the hydrodynamic diameter of the particles in the nanofluid was measured at a constant concentration of 500 ppm Si-NH_2_ nanoparticles with and without 1200 ppm Azarboo surfactant in distilled water. The results demonstrated two narrow bell-shaped size distributions ranging from 20 nm to 140 nm (Figure 8b). The average size of 57 nm was measured for Si-NH_2_, being higher than that of bare SiO_2_. The size was increased by about 5 nm and reached 62 nm after treatment with surfactant. The size change is due to the linkage of the Azarboo surfactant. Taking negative charges on the Azarboo surfactant into account, it could be conjugated on the surface of positively charged Si-NH_2_ nanoparticles using electrostatic forces. Thus, Si-NH_2_-surfactant nanoparticles would have a larger particle size, which is consistent with other studies [59].

The presence of Chooback surfactant in the structure of SiO_2_ nanoparticles was further studied by measuring the BET surface area before and after treatment with amine and surfactant. As shown in Figure 8c, bare SiO_2_ nanoparticles had a higher BET surface area than aminated and surfactant-modified nanoparticles (541 m^2^/g versus 519 m^2^/g and 425 m^2^/g). Chooback molecules could be adsorbed on SiO_2_ nanoparticles and modify their surfaces [15,61]. This result was confirmed using SEM images with a scale of 10 nm (Figure 9). SiO_2_ nanoparticles before treatment were round and spherical, with an average size of almost 5–10 nm (Figure 9a). However, after treatment with Chooback, they became foamy, whiter, and larger (Figure 9c), which confirms the surface modification. In contrast, no sensible change was observed in the form of SiO_2_ nanoparticles after amination, and they became only a bit whiter (Figure 9b).

### 3.2. The Effect of the Nanoparticle on Wettability

The natural surfactant of Azarboo and amine molecules were utilized for the surface modification of SiO_2_ nanoparticles. The modification was proven using different characterization tests. Herein, the performance of the developed nanofluid is studied for wettability alteration using spontaneous imbibition and surface imaging technique.

#### 3.2.1. Spontaneous Imbibition

Spontaneous imbibition occurs when a wetting fluid displaces a non-wetting fluid in porous media without external forces [62,63]. Three oil-wet carbonate samples were employed to know how Si-NH_2_ and Si-NH_2_-surfactant would affect the spontaneous imbibition oil recovery. Figure 10 demonstrates the oil recovery results for the samples after almost two months. As can be seen, the oil-wet sample had the lowest imbibition with an oil recovery of 14%, proving its oil-wet tendency. The amount of oil produced after modification with Si-NH_2_ was around 23%, and after treatment with Si-NH_2_-surfactant was about 28%. It was evident that almost 14% and 9% of the oil recovered should have been due to Si-NH_2_-surfactant and Si-NH_2_ nanoparticles. Even though the rate of oil production was noticeably higher when Si-NH_2_-surfactant was applied. So, this test is evidence of wettability alteration by the modified chemicals.

#### 3.2.2. Surface Imaging

In the previous section, both Si-NH_2_-surfactant and Si-NH_2_ nanoparticles were hydrophilic, but different oil recoveries were obtained. A surface imaging technique was employed to visualize the alterations originating from nanoparticle obstruction. Each of the core plugs was cut horizontally and split in two. Then, SEM photographs were taken before and after exposure to Si-NH_2_-surfactant and Si-NH_2_ nanofluids (Figure 11).

Figure 11a illustrates the morphology of the oleophilic media before treatment with nanofluid, and Figure 11b,c shows the oil-wet slice morphology after treatment with Si-NH_2_ and Si-NH_2_-surfactant, respectively. The rock sample soaked in oil (Figure 11a) has roughened a little after exposure to Si-NH_2_ nanofluid, as shown in Figure 11b. This roughness is because of the low-affinity adsorption of Si-NH_2_ nanoparticles on the carbonate surface. In contrast, Figure 11a,c showed that Si-NH_2_-surfactant nanoparticles had substantially adsorbed over the cleaned porous medium. Compared to treatment with Si-NH_2_, the open and visible pores had become closed and invisible, offering Si-NH_2_-surfactant nanoparticles further interacting with the oil-wet carbonate samples.

Surface roughness is a critical factor that affects wettability. Contact angles were also measured to confirm the nanoparticles’ adsorption. Angles were recorded after aging rock surfaces in oil, Si-NH_2_, and Si-NH_2_-surfactant nanofluids (Figure 12a). The contact angle for the oil-wet rock chip was 147°, confirming an initial oleophilic condition. Contact angles for Si-NH_2_-surfactant and Si-NH_2_ nanofluids were changed to 36° and 85°, respectively. Hence, amine-treated nanoparticles changed the wetting state to neutral-wet and surfactant-treated nanoparticles to strongly water-wet.

Rostami [64] stated the hydrophilic property of silica nanoparticles as the reason for changing the wetting condition to water-wetness. Khoramian et al. [65,66] showed the amphiphilic nature of graphene oxide nanosheets for restoring wettability to mixed-wetness. Thus, wettability alteration to a more water-wet state and change in surface roughness are because of the attraction between heavy oil compositions deposited on the rock surface and the hydrophobic tail of the natural surfactant in the Si-NH_2_-surfactant (Figure 12b). It, in turn, contributed to higher adsorption of Si-NH_2_-surfactant nanoparticles and water-wet wettability alteration.

The scaling approach was utilized to magnify the difference in imbibition oil recovery data [67]. Figure 13 is re-plotted where $R_{D}=\frac{R\left(t\right)}{R_{T}}$ and *R*(*t*) is oil recovery at different times, and *R_T_* is the final oil recovery. Hence, the *x*-axis shows imbibition duration, and the *y*-axis presents the normalized imbibition oil recovery of the core samples. From the results, the surfactant-treated sample (green line) exhibited a very swift imbibition process and reached maximum oil recovery after only nine days due to its hydrophilic nature (θ = 36°). In contrast, the non-treated oleophilic sample (black line) showed the slowest imbibition rate within twenty-eight days and the highest imbibition resistance. Capillary forces prevent the non-wetting fluid from imbibing in an oil-wet sample, slowing down the imbibition rate [68,69]. The Si-NH_2_ treatment could also restore the original wettability and increase the speed of spontaneous imbibition. However, Si-NH_2_ treatment permitted a considerable restoration of the rock sample wettability (θ = 85°). The results demonstrated the better effectiveness of the Si-NH_2_-surfactant in modifying the wettability and accelerating the process to reach the maximum recovery, nine days versus fifteen days. The results can be clarified more sensibly when they are made dimensionless using the spontaneous imbibition scaling parameters [70,71].

Herein, the scaling group of Mattax and Kyte [31] was utilized to calculate t_D_ based on the parameters listed in Table 4 for analytical comparisons of the results.

To estimate α, dimensionless imbibition oil recovery was plotted versus dimensionless time for all tests based on the model developed by Ma et al. [32] (Figure 14). Mattax and Kyte [31] showed that a constant production decline of 0.05 is devoted to strongly water-wet systems. If α < 0.05, then the system becomes less water wet. In our study, the oil-wet sample α is 0.002, which means the least water-wet condition, as expected. The imbibition results of the rock sample treated with Si-NH_2_ nanoparticles were fitted with α = 0.006, which showed a partially water-wet condition. Results of the case of the Si-NH_2_-surfactant nanoparticles matched the decline constant of 0.03, which shows a strongly water-wet porous media [72].

Three different alpha values indicated three different imbibition rates and wettability types. A not strongly water-wet system (Si-NH_2_) exhibited spontaneous imbibition but with lower imbibition rates. The rate of wettability alteration for surfactant-treated nanofluid was faster, and the process happened sooner. In conclusion, the proposed natural surfactant-based nanofluid can be promising for EOR operations due to higher and faster recovery rates. Thus, it was used for oil displacement and present core flooding.

### 3.3. Core Flooding

Samples 3 and 4 were soaked in oil and made oleophilic for core flooding experiments. The secondary water flooding was done by injecting six pore volumes of brine with 180,000 NaCl. As shown in Figure 15, only 42 and 45% of the oil was recovered by water flooding. In other words, more than half of the original oil in place was left intact inside the core samples. This low and unfavorable oil recovery was anticipated to the oil-wetness of the rock samples. It was persuasive enough to inject a one-pore volume of Si-NH_2_-surfactant and Si-NH_2_ nanofluids into cores No. 3 and 4 and allow them to be exposed to the nanoparticles for 24 h. This treatment was done to modify the wettability of the cores toward less oil-wetness. The cores were then saturated with oil until S_wir_ was attained. When the 24-h nano treatment was finished, the second brine flooding was conducted, and the volume of oil recovered was recorded (Figure 15). The second flooding resulted in an oil recovery of 59% for Si-NH_2_-surfactant treatment and 49% for Si-NH_2_ treatment, showing a surpassing effect of Si-NH_2_-surfactant nanoparticles on enhancing oil recovery compared with Si-NH_2_. The improvements in oil recovery is due to the wettability restoration and IFT reduction caused by the nanoparticles. The presence of surfactant-treated nanoparticles in the base fluid led to a decrease in interfacial tension from 32 to 20 dyne/cm. This result, which is in a good agreement with previous studies [73], is due to the connection between the hydrophobic head of Azarboo surfactant and crude oil molecules [74].

The primary water flooding was run when capillary pressure was negatively high due to the oil-wet inclination of the porous media. Therefore, only wide pores were depleted by water, and a considerable value of oil was trapped inside narrow pores. Aging by the nanoparticles led to the moderate and strong adsorption of Si-NH_2_ and Si-NH_2_-surfactant on the carbonate media pores and throats, restoring the wettability and decreasing the negative capillary pressure. It, in turn, reinforced water suction into the narrow and small pores and promoted oil recovery to different degrees.

## 4. Conclusions

A non-toxic anionic surfactant, Azarboo, was obtained from the bony roots of Acanthophyllum for possible EOR applications. It was conjugated to positively charged amine-treated SiO_2_ nanoparticles and characterized using FT-IR, Zeta potential, DLS, BET, and SEM analyses. Then, the effects of Si-NH_2_ and Si-NH_2_-surfactant nanoparticles on the spontaneous imbibition of strongly oil-wet carbonate rocks were experimentally and theoretically examined.

Imbibition results proved the active role of Si-NH_2_ and Si-NH_2_-surfactant nanoparticles. Maximum oil recovery for treatment with Si-NH_2_-surfactant and Si-NH_2_ was achieved after nine and fifteen days, respectively, while this result for an oil-wet sample was obtained after twenty-eight days. The rate of wettability alteration was found faster by silica and surfactant together, which was supported by SEM images.

The spontaneous imbibition data were scaled using an analytical model. A decline production constant of 0.006 for the Si-NH_2_ imbibition test confirmed that it acted like a partially water-wet system. In contrast, the Si-NH_2_-surfactant imbibition test proved a nearly strongly water-wet system with a decline production constant of 0.03. The hydrophobic tails of the natural surfactant could link to oil compositions deposited on porous media and speed up oil production by more wettability alteration and IFT reduction.

The results of core flooding experiments showed the effectiveness of Si-NH_2_ and Si-NH_2_-surfactant nanoparticles for EOR purposes. The oil production rate experienced an increase of about 15% for Si-NH_2_-surfactant nanofluid and almost 7% for Si-NH_2_ nanofluid. Overall, the hybrid application of the natural surfactant and silica nanoparticles could improve oil production more than the nano-treatment with anime molecules.

## Figures and Tables

**Figure 1 nanomaterials-12-03563-f001:**
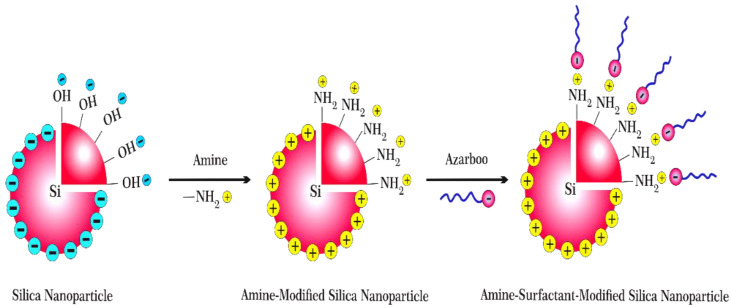
Amine and surfactant modification of silica nanoparticles using Azarboo and amine molecules.

**Figure 2 nanomaterials-12-03563-f002:**
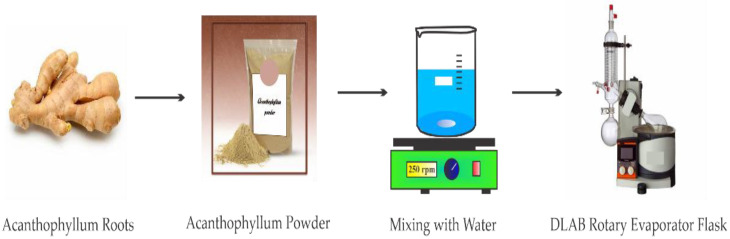
Experimental procedures used in preparing the Azarboo Surfactant.

**Figure 3 nanomaterials-12-03563-f003:**
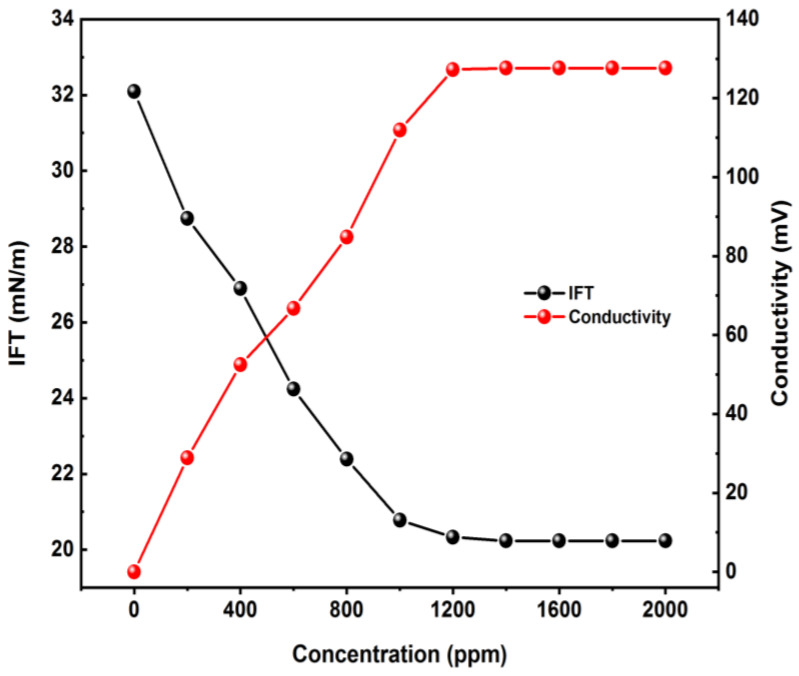
Surface conductivity and IFT versus Azarboo concentration for CMC determination.

**Figure 4 nanomaterials-12-03563-f004:**
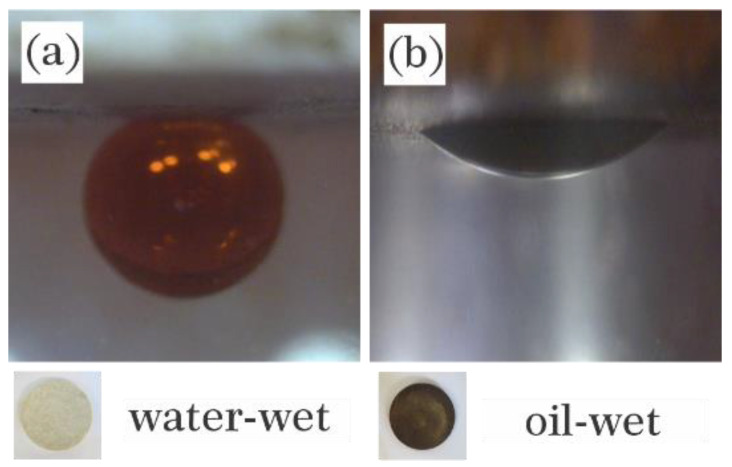
Images of carbonate thin sections (**a**) before and (**b**) after aging in crude oil correspond contact angles.

**Figure 5 nanomaterials-12-03563-f005:**
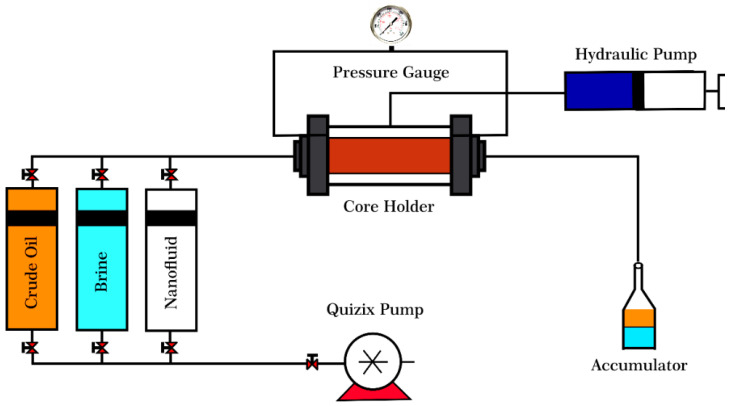
A schematic illustration of the core flooding setup.

**Figure 6 nanomaterials-12-03563-f006:**
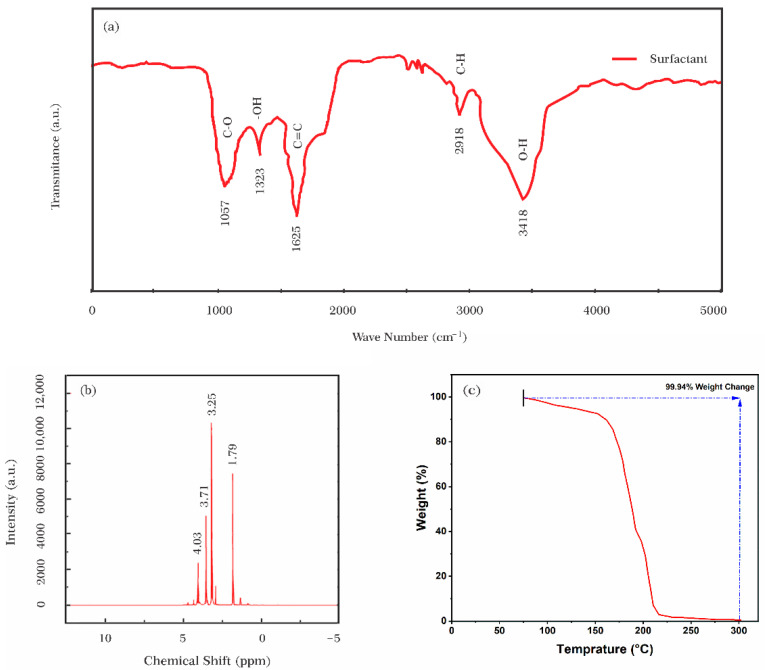
(**a**) FT-IR spectrum (**b**) ^1^H-NMR spectrum and (**c**) TGA analysis of Azarboo surfactant.

**Figure 7 nanomaterials-12-03563-f007:**
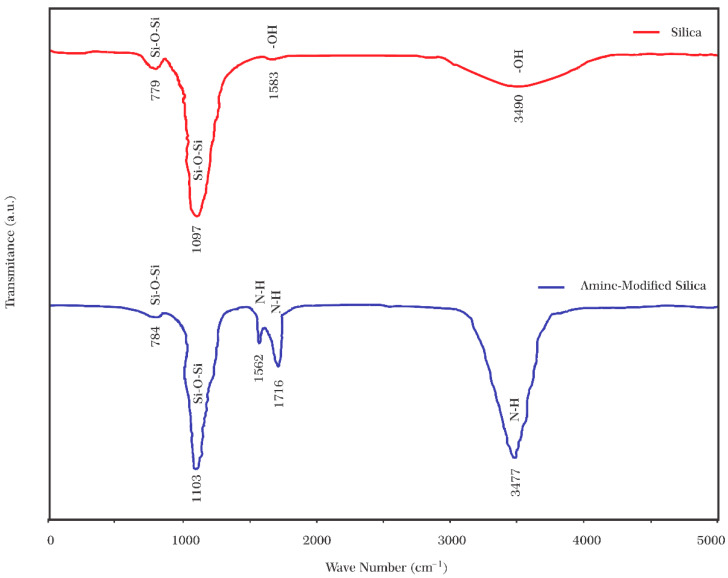
FT-IR spectrums of bare and amine-modified SiO_2_ nanoparticles.

**Figure 8 nanomaterials-12-03563-f008:**
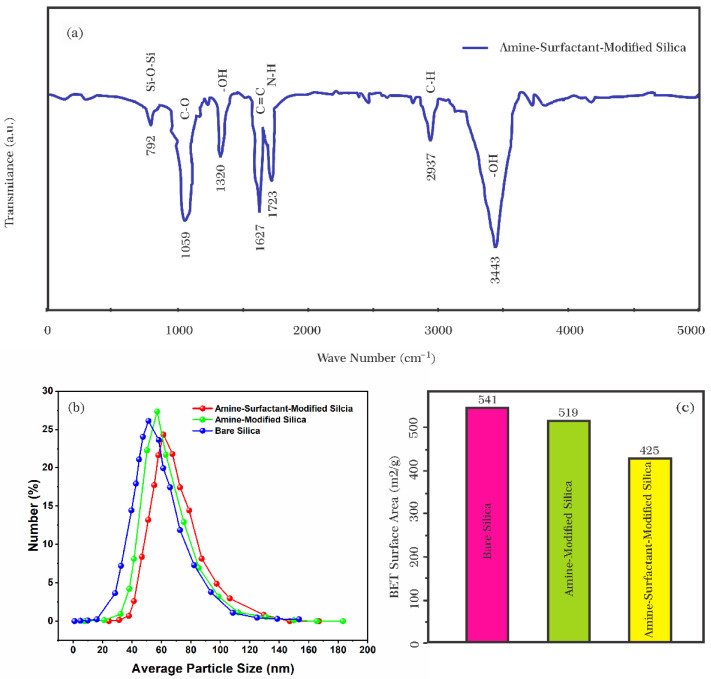
(**a**) FT-IR spectrum of amine-surfactant-modified SiO_2_ nanoparticles, (**b**) particle size distribution and (**c**) BET surface area of bare, amine-treated, and amine-surfactant-treated SiO_2_ nanoparticles.

**Figure 9 nanomaterials-12-03563-f009:**
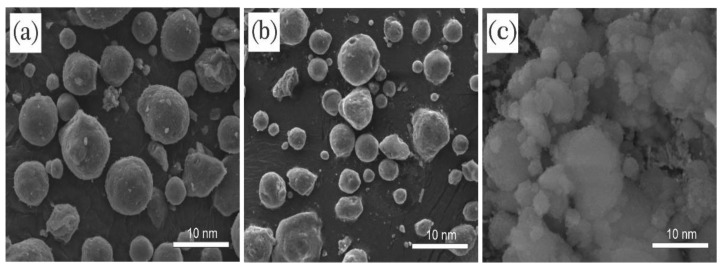
SEM images of (**a**) bare, (**b**) aminated, and (**c**) amine-surfactant-treated silica nanoparticles with a 10-nm scale.

**Figure 10 nanomaterials-12-03563-f010:**
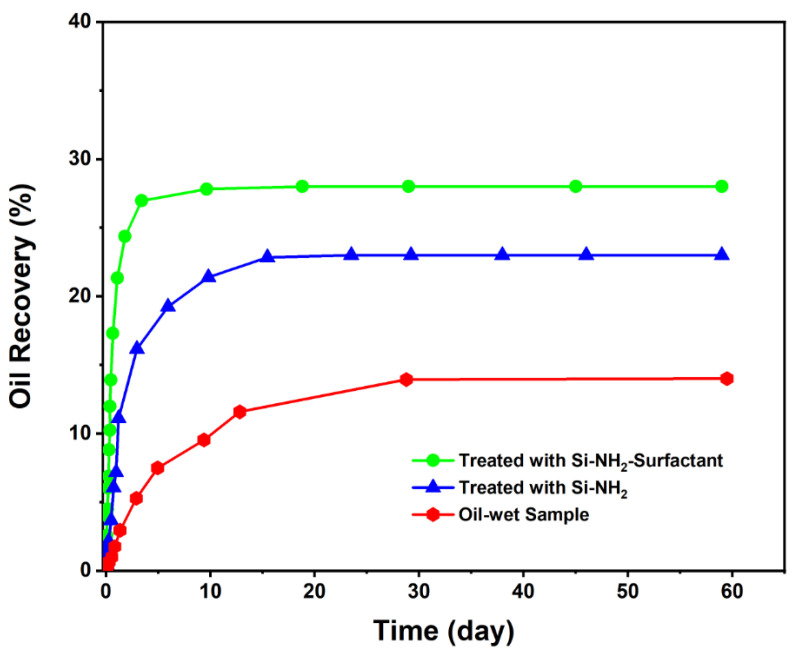
Spontaneous imbibition oil recoveries of the oil-wet sample, and core plugs treated with Si-NH_2_-surfactant and Si-NH_2_ nanofluids.

**Figure 11 nanomaterials-12-03563-f011:**
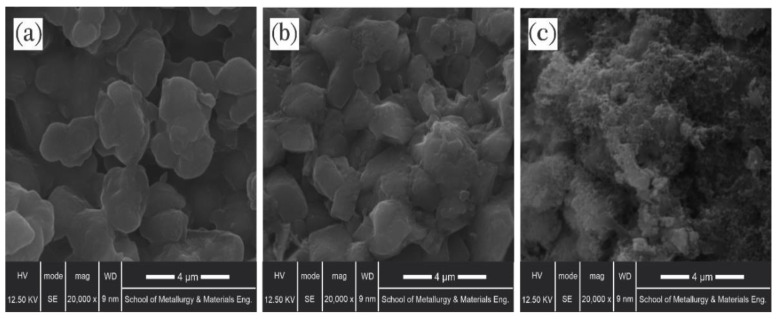
SEM images of the rock samples, (**a**) soaked in crude oil, (**b**) treated with Si-NH_2_, and (**c**) drenched in Si-NH_2_-surfactant.

**Figure 12 nanomaterials-12-03563-f012:**
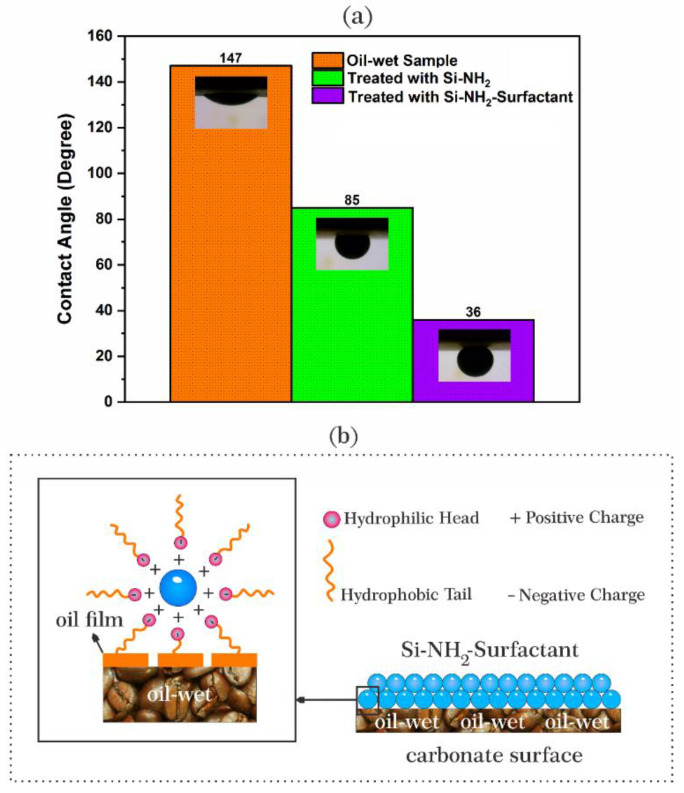
(**a**) The wetting angle (θ) values for three carbonate rock samples and (**b**) the representational image of wettability alteration by Si-NH_2_-Surfactant nanoparticles.

**Figure 13 nanomaterials-12-03563-f013:**
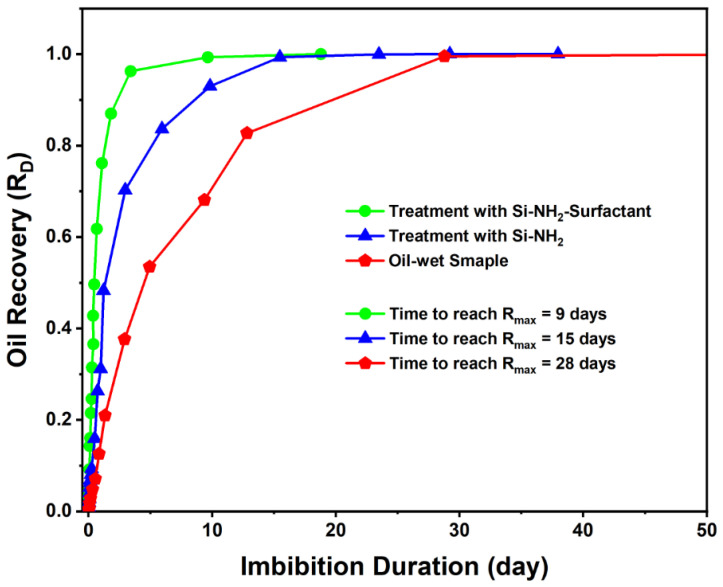
Normalized spontaneous imbibition curves for the three tests.

**Figure 14 nanomaterials-12-03563-f014:**
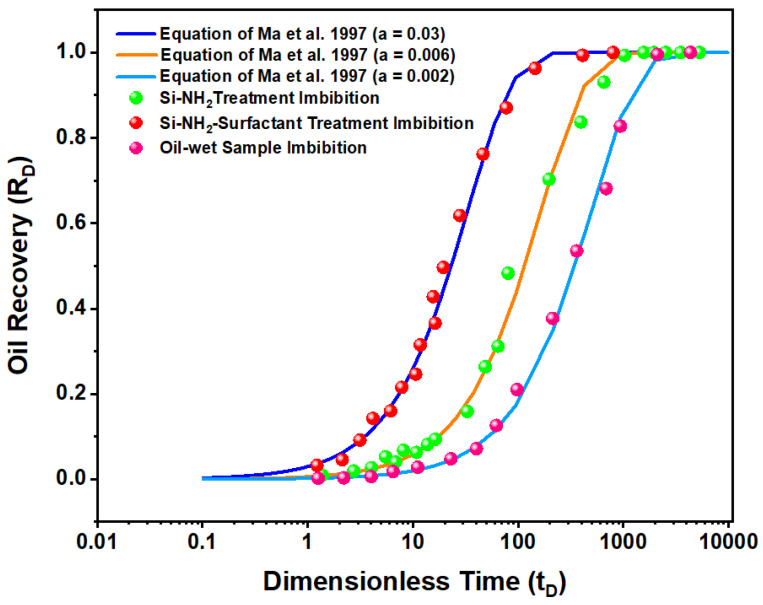
Imbibition oil recovery versus dimensionless time for the oil-wet sample and samples treated with Si-NH_2_-surfactant and Si-NH_2_ nanofluids. All tests are based on the model developed by Ma et al. 1997.

**Figure 15 nanomaterials-12-03563-f015:**
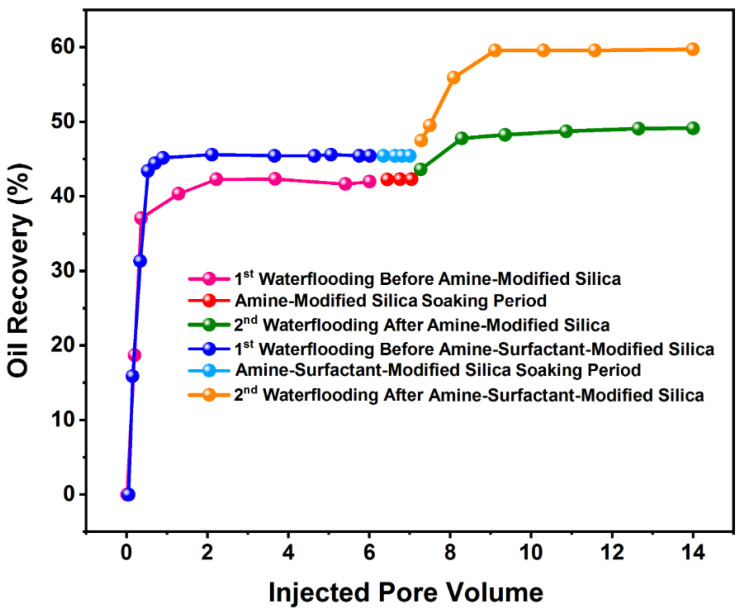
Oil recovery values before and after treatment with Si-NH_2_ and Si-NH_2_-surfactant nanofluids.

**Table 1 nanomaterials-12-03563-t001:** The physical and chemical properties of the silica nanoparticles.

Nanoparticle	Color	Shape	Average Size	pH	Density (g/cc)	Surface Area (m^2^/g)
SiO_2_	White	Spherical	5–15	3.7–4.7	5 × 10^−2^	200

**Table 2 nanomaterials-12-03563-t002:** The physical characteristics of the carbonate samples utilized in this survey.

Core No.	Permeability	Porosity	Diameter	Length	Pore Volume	S_wir_ (Irreducible Water Saturation)
1	48.3 mD	21.6%	3.82 cm	6.72 cm	16.6 cc	29.5
2	52.7 mD	19.2%	3.87 cm	6.39 cm	14.4 cc	28.3
3	54.7 mD	19.8%	3.81 cm	6.48 cm	14.6 cc	26.3
4	49.3 mD	18.5%	3.85 cm	6.53 cm	14.1 cc	30.1

**Table 3 nanomaterials-12-03563-t003:** The oil properties and composition at 14.7 psi and 60° F.

Chemical Properties	Value
C_1_	0.08 mole%
C_2_	0.14 mole%
C_3_	1.48 mole%
iC_4_	1.06 mole%
nC_4_	4.65 mole%
iC_5_	2.69 mole%
nC_5_	1.29 mole%
C_6_	8.23 mole%
C_7+_	80.38 mole%
Gravity	36.8° API
Density	0.823 g/cc
Viscosity	23.9 cp

**Table 4 nanomaterials-12-03563-t004:** The characteristics of the carbonate samples and fluids used in spontaneous imbibition experiments.

Parameter	Oil-Wet Sample	Aminated Sample	Surfactant-Amine-Treated Sample
Permeability (mD)	52.7	48.3	52.7
Porosity (%)	19.2	21.6	19.2
Length (cm)	6.39	6.72	6.39
Water Viscosity (cp)	0.97	1.07	1.03
Oil Viscosity (cp)	23.9	23.9	23.9
Interfacial Tension (dyne/cm)	32	20	30
$t_{D}=\left(\frac{0.00031415}{L_{C^{2}}}\sqrt{\frac{k}{\varphi }}\frac{\sigma_{\mathrm{ow}}}{\sqrt{\mu_{o}\mu_{w}}}\right)t$	t_D_ = 0.051 t (hr.)	t_D_ = 0.025 t (hr.)	t_D_ = 0.046 t (hr.)

## Data Availability

The data will be available on request.

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
