# Peer review of "Spontaneous Imbibition Oil Recovery by Natural Surfactant/Nanofluid: An Experimental and Theoretical Study"

_nanomaterials, 2022, doi:10.3390/nano12203563_

Round 1

Reviewer 1 Report

In this paper an anionic surfactant was proposed and the mechanism of EOR was discussed. I think authors have done a solid work and this paper can be considered for publication in Nanomaterials. Some suggestions and questions are listed as follows,

1. In Fig. 8, why does have oil recovery from oil-wet sample? The sample may be mixed wettability.

2.It is good to observe the higher IFT and the improved wettability alteration. It made a good contribution to spontaneous imbibition. However, the higher IFT may increase the residual oil saturation for water flooding. In this paper, the surfactant can improve oil recovery by water flooding. It is interesting and good. Can authors comment more about mechanism of this?

3. Do authors have applied a patent?

3. Some paper may make a contribution to this work.

Surfactant-Enhanced Oil Recovery from Fractured Oil-Wet Carbonates: Effects of Low IFT and Wettability Alteration

A critical review on fundamental mechanisms of spontaneous imbibition and the impact of boundary condition, fluid viscosity and wettability

Experimental Investigation of In-Situ Emulsion Formation To Improve Viscous-Oil Recovery in Steam-Injection Process Assisted by Viscosity Reducer

Author Response

Reviewer #1:

In this paper, an anionic surfactant was proposed and the mechanism of EOR was discussed. I think the authors have done a solid work, and this paper can be considered for publication in Nanomaterials. Some suggestions and questions are listed as follows:

  • Before considering the concerns, we thank the reviewer warmly for giving positive feedback on our experimental study.

Comments:

In Fig. 8, why does have oil recovery from oil-wet sample? The sample may be mixed wettability.

Answer:

Thanks for the comment. To achieve the strong oil-wet state, core plugs were all aged in crude oil for almost three weeks to before imbibition experiments. This fact was mentioned in a new paragraph (2.2.5. Oil-Wet Procedure) with a new figure (Figure 4) .

It is good to observe the higher IFT and the improved wettability alteration. It made a good contribution to spontaneous imbibition. However, the higher IFT may increase the residual oil saturation for water flooding. In this paper, the surfactant can improve oil recovery by water flooding. It is interesting and good. Can authors comment more about the mechanism of this?

Answer:

In this study, IFT enhancement after using the natural surfactant of Chooback did not result in a reduction in residual oil saturation. The effect of surfactant concentration on IFT reduction was studied and an optimum concentration of 1200 ppm was found. Measurements showed an almost 39% decrease in IFT, confirming a very good influence on emulsification and IFT reduction. A new paragraph was added to 2.2.2. Natural Surfactant Extraction and Optimization.  

As for the associated mechanism, the core samples were aged in surfactant-treated nanofluids for 24 hours. Surfactants and nanoparticles could change the surface wettability of core plugs and decrease IFT when exposed to crude oil. That is why waterflooding could improve oil recovery. This mechanism was explained in section 3.3.

Do authors have applied for a patent?

Answer:

No, we have not applied for a patent yet.

Some papers may make a contribution to this work:

Paper 1) Surfactant-Enhanced Oil Recovery from Fractured Oil-Wet Carbonates: Effects of Low IFT and Wettability Alteration.

Paper 2) A critical review on fundamental mechanisms of spontaneous imbibition and the impact of boundary conditions, fluid viscosity, and wettability.

Paper 3) Experimental Investigation of In-Situ Emulsion Formation to Improve Viscous-Oil Recovery in Steam-Injection Process Assisted by Viscosity Reducer.

Answer:

The recommended papers were all studied and utilized in the manuscript to have an up-to-date literature review. They were used as references [49], [48], and [5], respectively.  

Reviewer 2 Report

The main idea of the work is rather worthy, a practical potential of the material is of a certain interest. In my opinion, the work can be published upon some minor but very important corrections.

1.       The Experimental section must be supplemented and re-organized in order to provide the information on the sources of all the materials as well as the description of all the synthetic and analytical methods used.

2.       Please provide the information on the chemical composition and structure of the surfactant used.

3.       Throughout the manuscript, please provide comparable concentration values for all the materials. For example, wt.% of nanoparticles are hardly to be compared to ppm of surfactant.

4.       Please, provide the SEM images of amino-modified silica microparticles. 10 nm and 20 nm are not magnifications.

5.       In Fig. 6, please provide particle size distributions and BET data for bare silica.

6.       Please provide the quantitative information on the chemical composition (water content, aminogroups content, surfactant content) for the silica microparticles, bare and modified by different techniques.

7.       Please re-check the number of significant digits for the numerical values provided in the manuscript. In some cases they are very confusing.

8.       The English requires extensive polishing throughout the manuscript.

Author Response

Reviewer #2:

The main idea of the work is rather worthy, a practical potential of the material is of a certain interest. In my opinion, the work can be published upon some minor but very important corrections.

  • We would like to thank the reviewer gratefully for considering our research worthy before regarding the below comments.

The Experimental section must be supplemented and reorganized in order to provide information on the sources of all the materials as well as the description of all the synthetic and analytical methods used.

Answer:

As requested, a new section “2. Experimental Section” was supplemented and its subsections were reorganized.  In 2.1. Materials, all sources of materials including nanoparticles, the natural surfactant, crude oil, rock samples, and amination chemicals are discussed, and in 2.2. Methodology, all approaches used for fluids preparation were added.  Concerning the natural surfactant synthesis, a new paragraph with a new figure (Figure 2) showing the natural surfactant extraction procedure was added to section 2.2.2. For the nanofluids preparation, there have already been different sections like 2.2.1. and 2.2.3. Also, in 2.2.6. (Spontaneous Imbibition) the concepts of the analytical model used in this study were explained in more details.  

Please provide information on the chemical composition and structure of the surfactant used.

Answer:

As requested, the functional groups and chemical compositions of the natural surfactant were studied by FT-IR, 1H-NMR, and TGA analyses. These analyses were provided in Figures 6a-b and Figure 8 (red line). The results were sufficiently discussed in 3.1 Characterization Results (3.1.1. Natural Surfactant). The procedure of the analyses was also explained in 2.2.4 Surfactant Characterization. 

Throughout the manuscript, please provide comparable concentration values for all the materials. For example, wt.% of nanoparticles are hardly to be compared to ppm of surfactant.

Answer:

As recommended, concentration values throughout the manuscript were changed from wt.% to ppm.

Please, provide the SEM images of amino-modified silica microparticles. 10 nm and 20 nm are not magnifications.

Answer:

As suggested, the SEM image of amino-modified silica microparticles was added (Figure 10b) and compared with bare and surfactant-modified silica nanoparticles. The word “magnifications” in the caption of SEM images and in the text was modified to “scale”.

In Fig. 6, please provide particle size distributions and BET data for bare silica.

Answer:

As requested, both particle size distribution and BET data for bare silica nanoparticles were provided and added to 3.1.3. Amino-Surfactant-Modified Nanoparticles in Figure 9.  

Please provide the quantitative information on the chemical composition (water content, amino groups content, surfactant content) for the silica microparticles, bare and modified by different techniques.

Answer:

In this study, we used 1200 ppm of natural surfactant and 500 ppm of silica nanoparticles for imbibition and flooding experiments. Unfortunately, no results are available regarding quantitative information on water content, amino content, and surfactant content.

Please re-check the number of significant digits for the numerical values provided in the manuscript. In some cases, they are very confusing.

Answer:

Fixed

The English require extensive polishing throughout the manuscript.

Answer:

As recommended by the reviewer, the English of the manuscript was revised and improved.

Reviewer 3 Report

What a great paper with novel findings! Methodology is detailed and clear, with a range of appropriate techniques employed to produce a well rounded set of interfacial properties and characterise EOR behaviour. Results are consistently presented and suitably discussed. I would suggest to make slight changes to the introduction/lit. review section though.  It would be good to focus the introduction to reflect on the work done so far on amine-modified silica, and specifically Azarboo (Chooback). Please avoid grouped references, for example [3-5].

Author Response

Reviewer #3:

What a great paper with novel findings! The methodology is detailed and clear, with a range of appropriate techniques employed to produce a well-rounded set of interfacial properties and characterize EOR behavior. Results are consistently presented and suitably discussed.

  • Before moving forward to consider the comments, we are deeply grateful for the kind support of the reviewer.

I would suggest making slight changes to the introduction or literature review section.  It would be good to focus the introduction to reflect on the work done so far on amine-modified silica, and specifically Azarboo (Chooback).

Answer 1:

As suggested, two separate paragraphs were added to briefly cover recent papers on natural surfactants (five cases) and the amination of nanoparticles (three cases).  These changes can be seen in 1. Introduction.  

Please avoid grouped references, for example [3-5].

Answer 2:

Fixed.